# Validity of Bioimpedance Equations to Evaluate Fat-Free Mass and Muscle Mass in Severely Malnourished Anorectic Patients

**DOI:** 10.3390/jcm9113664

**Published:** 2020-11-14

**Authors:** Moise Coëffier, Mathilde Gâté, Agnès Rimbert, André Petit, Vanessa Folope, Sébastien Grigioni, Pierre Déchelotte, Najate Achamrah

**Affiliations:** 1Nutrition Department, Rouen University Hospital Center, 76000 Rouen, France; moise.coeffier@univ-rouen.fr (M.C.); mathilde.gate@chu-rouen.fr (M.G.); agnes.rimbert@chu-rouen.fr (A.R.); andre.petit@chu-rouen.fr (A.P.); vanessa.folope@chu-rouen.fr (V.F.); sebastien.grigioni@chu-rouen.fr (S.G.); pierre.dechelotte@inserm.fr (P.D.); 2Nutrition, Inflammation et Dysfonction de l’axe Intestin-Cerveau, Institute of Research and Innovation in Biomedicine (IRIB), UNIROUEN, Normandie University, INSERM UMR 1073, 76000 Rouen, France; 3Clinical Investigation Centre CIC 1404, INSERM and Rouen University Hospital, 76000 Rouen, France

**Keywords:** anorexia nervosa, body composition, dual X ray absorptiometry (DXA), Bioelectrical impedance analysis (BIA), predictive equations

## Abstract

Background: Bioelectrical impedance analysis (BIA) is a simple and rapid technique to measure body composition (BC). Validity of BIA in patients with low body mass index (BMI) remains controversial. We assessed the validity of several BIA equations to evaluate fat-free mass (FFM), fat mass (FM) and muscle mass in patients with anorexia nervosa (AN) by using dual X ray absorptiometry (DXA) as reference. Methods: Sixteen BIA equations developed for FFM and appendicular lean mass (ALM) were applied on electrical data measured by BIA in AN patients with BMI <16 kg/m². BIA and DXA were done the same day after overnight fasting. Results were compared with the Bland–Altman method, Pearson correlation and a Lin concordance test. Results: Data from 115 female AN patients (14.6 ± 1.2 kg/m^²^; 32.3 ± 14.5 years) were included. FM and FFM assessed by DXA were, respectively, 4.2 ± 2.4 kg and 35.5 ± 3.8 kg. The best results were obtained with Sun’s equation: respectively for FM and FFM, Bland Altman bias at 0.548 and 0.706 kg, Pearson correlation r at 0.86 and 0.86 and Lin concordance coefficient at 0.81 and 0.84. However, confidence intervals (CI) at 95% were high (−2.73–3.83 kg for FM; −4.55–3.13 kg for FFM). Other equations also showed high 95% CI. Accuracy was acceptable for Sun and Bedogni equations for FFM (approximately 66%) but very low for FM prediction considering all equations (<15%). Concerning ALM evaluated at 14.88 ± 2.04 kg by DXA, only Scafoglieri and Yoshida equations showed acceptable values: bias (−0.2 and 2.8%), Pearson r (0.89 and 0.86), Lin concordance coefficient (0.82 and 0.82) and accuracy (83.5 and 82.6%). Confidence intervals at 95% were high for both equations (−2.1–2.0 for Scafoglieri equation and −1.6–2.4 for Yoshida equation). Conclusion: In AN patients with BMI < 16 kg/m², no BIA equation tested was adapted to evaluate BC at the individual level.

## 1. Introduction

Anorexia nervosa (AN) is an eating disorder defined by undernutrition, i.e., body mass index (BMI) <18.5kg/m² and intense fear of gaining weight [1], and is associated with a decrease of fat mass (FM) and fat-free mass (FFM) [2]. The assessment of body composition plays a key role in evaluating nutritional status in AN because body weight and BMI could be unreliable in cases of edema for example. Several accurate techniques for the assessment of body composition in humans have been developed [3].

The reference method is dual-energy X-ray absorptiometry (DXA), which provides a rapid assessment of FM, FFM and bone mineral density [4], but DXA is expensive and requires specialized radiology equipment and environment, thus it is hardly feasible in routine clinical practice. Inversely, bioelectrical impedance analysis (BIA) is easily used for the assessment of body composition in clinical practice and research studies. Indeed, BIA is a non-invasive, simple, low-cost device that estimates total body water through the body’s resistance to a small alternating current [5]. Several BIA devices are available and use manufacturers’ predictive equations involving the body’s resistance to current flow, and other variables such as body weight, height and age. BIA equations developed in a specific population are only generalizable to similar populations, so caution is needed when applying them to a different population, in order to avoid imprecise results and wrong interpretation.

Comparison of body composition assessment by DXA and BIA in AN patients has been reported in some studies [6,7,8,9] showing inconsistent results. We previously reported in malnourished patients that BIA underestimated FFM and overestimated FM when using the manufacturer’s BIA equation [10]. Interestingly, higher differences were observed for lower BMIs. The lack of a disease-specific equation developed for the estimation of body composition in severely malnourished patients may explain this result. 

Furthermore, strong evidence shows that undernutrition associated with loss of muscle mass negatively impacts clinical outcomes [11] and the preservation or improvement of muscle mass represents a challenge not only in AN but also in chronic diseases. Reduced muscle mass is a phenotypic criterion for malnutrition in GLIM (Global Leadership Initiative on Malnutrition) criteria [12] and in the recommendations from the European Working Group on Sarcopenia in Older People [13]. However, to our knowledge, no study has evaluated the validity of equations developed to estimate muscle mass or appendicular lean mass (ALM) in anorectic patients.

In this context, we aimed to evaluate the validity of BIA equations developed either for FFM or for muscle mass in a large sample of severe patients with AN (BMI <16 kg/m^2^) by using DXA as reference.

## 2. Materials and Methods

### 2.1. Patients

Patients were included from the Department of Clinical Nutrition (Rouen University Hospital, Rouen, France) from 2010 to 2016. Inclusion criteria were: female sex, age above 18 years, without acute disease, followed for anorexia nervosa according to the DSM-IV classification [1] (Appendix A).

The same operator measured weight and height dressed in light clothes without shoes, after an overnight fasting period of 12 h. Body weight (kg) and height (m²) were used to calculate BMI (kg/m²). On the same day, body composition assessment, using both DXA and BIA, was performed for all patients, in a row, in the same conditions. The study was approved by the Local Ethics Committee for Non-Interventional Studies (CERNI, Comité d’Ethique pour la Recherche Non Interventionnelle), patients gave their consent and all data were fully anonymized. 

### 2.2. Dual-Energy X-ray Absorptiometry (DXA)

DXA was performed using a Lunar Prodigy Advance (General Electric Healthcare, New York, USA) on the whole body, without specific preparation. QA and QC data evaluation was done every morning when patient assessment was planned and at least 3 days per week. QA and QC data were independently monitored every month. Over the 6-year period, we observed no deviation and there were no firmware or software upgrades. The DXA equipment was serviced by the manufacturer at least once per year. All patients were measured in underwear, without metal accessories worn. DXA uses two X-ray beams with different energy levels. Based on their X-ray attenuation properties, FFM, i.e., lean mass and bone mineral content, and FM were measured.

### 2.3. Bioelectrical Impedance Analysis (BIA)

Body composition, FFM and FM, was assessed using multifrequency BIA (Bodystat Quadscan 4000, Bodystat Ltd, UK) as previously described [14], according to the manufacturer’s recommendations. A calibration was done regularly (at least twice per year) by using a manufacturer’s calibrator measuring impedance at each frequency. BIA resistance and reactance were measured at 50 kHz, while BIA impedance was determined at each frequency.

Eight BIA equations (Table 1) were retrospectively applied to these electrical data measured by BIA to evaluate body composition [15,16,17,18,19,20,21,22]. Bedogni and Scalfi’s equations were developed in AN patients. Five of these equations were developed to calculate FFM while 3 equations calculated total body water. In this latter case, FFM was then calculated taking into account FFM hydration at 73%. FM was then calculated by the difference between body weight and FFM. We also applied 8 BIA equations (Table 2) previously developed to estimate skeletal muscle mass (SMM) or ALM [23,24,25,26,27,28,29,30,31]. 

### 2.4. Statistical Analysis 

Results obtained by DXA and BIA according to the equation used were compared by using the Bland–Altman method leading to the bias and the confidence interval at 95% (95% CI). For FFM and FM, the accuracy of each BIA equation was calculated by the percentage of patients with a difference between BIA and DXA results within ±5% of the data obtained by DXA. Pearson correlations were also performed and a *p*-value < 0.05 was considered significant. Then, the concordance coefficient of Lin was calculated taking into account that a ρ value ≥ 0.81 was considered fairly good as previously described [32].

## 3. Results

One hundred and fifteen patients were included in the study. Demographic data are displayed in Table 3. Briefly, patients were 32.3 ± 14.5 years old (mean ± standard deviation) and had a BMI of 14.6 ± 1.2 kg/m^2^. By using DXA, patients exhibited 4.2 ± 2.4 kg of FM (11.0 ± 5.6% of body weight), 35.5 ± 3.8 kg of FFM (89.0 ± 5.5%) including 14.88 ± 2.04 kg of ALM. 

Electrical data obtained by BIA (reactance, resistance and impedance) are shown in Table 4 and results of the Bland–Altman comparison, Pearson correlations and concordance coefficient of Lin are shown in Table 4 and Figure 1. 

### 3.1. Comparison of DXA and Derived BIA Values for Fat Mass and Fat-Free Mass

For FFM, Pearson correlation r was greater than 0.8 for all equations except for Scalfi-2 and the concordance coefficient of Lin was greater than 0.81 only for Sun and Bedogni equations (Figure 1). Among the tested BIA equations, four equations (Kyle, Sun, Bedogni and Scalfi-2) showed a bias of less than ±5%. Particularly, Sun, Bedogni and Scalfi-2 exhibited a bias of less than ±2% with a difference in FFM of 0.70, 0.34 and 0.52 kg, respectively. However, only the Sun and Bedogni equations showed acceptable accuracy around 66% (Table 5). Finally, all tested equations exhibited very large 95% confidence intervals (Figure 1 and Table 5). 

For FM, only the Sun, Roubenoff, Kushner and Scalfi-2 equations showed a Pearson correlation r greater than 0.8. In addition, the concordance coefficient of Lin was 0.81 for the Sun equation while it was less than 0.7 for all other equations (Figure 1) which is considered a mediocre result [33]. Even if the Sun, Bedogni and Scalfi-2 equations showed a difference in FM of less than 1 kg, there was no equation providing a bias of less than ±5%. In addition, accuracy was very low for all equations. Indeed, the Sun equation predicted accurate FM for 13.9% of patients whereas other equations showed accuracy in 0.9 to 5.2% of patients (Table 5). 

### 3.2. Comparison of DXA and Derived BIA Values for Appendicular Lean Mass

For ALM, all tested equations showed a Pearson correlation r greater than 0.81 except Kim’s equation (Table 6) but only two equations had a concordance coefficient of Lin considered fairly good, Scafoglieri and Yoshida equations (Figure 2 and Table 6). Among eight tested equations, four showed a bias of less than 10%; particularly, a bias of −0.2 and 2.8% for Scafoglieri and Yoshida equations, respectively. These two equations also provided the best accuracy (greater than 80%) (Table 6).

## 4. Discussion

In this study, we evaluated the validity of several BIA equations developed for FFM and muscle mass, in 115 severely malnourished patients with AN, by using DXA as reference. Regarding FM and FFM, the best results were obtained with the Sun equation reporting low bias (0.548 and 0.706 kg respectively), acceptable accuracy for FFM (66.9%), but poor accuracy for FM (13.9%). However, 95% CIs were high for all equations, highlighting the lack of a disease-specific equation developed for the estimation of body composition in AN patients. Previously, Birmingham et al. reported the inability of BIA to detect changes in body composition due to altered hydration, compared to skinfold measurements, in AN patients [9]. Skinfold measurements and DXA were similar in performance for the estimation of percentage of body fat in 80 AN patients [8]. Bonaccorsi et al. compared BIA (BIA software and Deurenberg equations) to DXA in 30 young girls (11 to 19 years old) with AN [6]. They found a high correlation between FFM values estimated with the two methods (BIA software vs. DXA r = 0.917, *p* < 0.001; Deurenberg equation vs. DXA r = 0.931, *p* < 0.001). However, the limits of agreement were high for FFM (±3.34 kg for the BIA software and ±2.96 kg for the Deurenberg equation) and FM (±4.60 kg for the BIA software and ±3.82 kg for the Deurenberg equation). In our study, the Deurenberg equation also provided high 95% CI. Mattar et al. reported that the Deurenberg equation gave the best estimates of FFM and FM, compared to DXA, in 50 AN patients (BMI = 14.3 ± 1.49 kg/m², age = 19.98 ± 5.68 years), while the Sun equation gave the broadest differences for FM and FFM [7]. Recently, Marra et al. observed in 82 AN patients that the five tested predictive BIA equations exhibited very low accuracy at population and individual levels, compared to DXA, with a percentage of accurate predictions varying from 12.2% to 35.4% with the Kyle and Sun equations respectively [34]. Differences in the BIA device used could explain the inconsistencies observed, as well as the characteristics of studied populations. In our study, AN patients had substantially lower BMI (BMI = 14.6 ± 1.2 kg/m² vs. 15.7 ± 1.7 kg/m²), higher FM (11.0 ± 5.6% vs. 9.8 ± 5.0%), and were older (32.3 ± 14.5 years vs. 20.5 ± 3.7 years). Other physiological (e.g., hydration) and anthropometric (e.g., segment lengths, circumferences and volumes) variables may also explain individual discrepancies.

To our knowledge, this is the first study to evaluate BIA equations developed for ALM in AN patients, by using DXA as reference. Scafoglieri and Yoshida equations showed acceptable results at the population level with low bias (−0.03 and 0.41 kg, respectively), and good accuracy (83.5 and 82.6%, respectively) but exhibited high 95% limits of agreement. Recently, Moore et al. reported that ALM estimates were equivalent between DXA and BIA, in 179 normal-weight adults [35], even if the 95% limits of agreement were high. In addition, the authors used the unknown manufacturer’s equation. The BIA equation tested does not seem to be adapted to severely malnourished patients with AN to evaluate muscle mass at the individual level. 

Both FFM and FM compartments are markedly altered during AN. The use of BIA allows the monitoring of compartmental weight gain during refeeding. A recent meta-analysis by Hübel et al. reported 50% less FM and 5 kg less FFM in AN patients compared to the control group [36]. Recovery was associated with restored FM primarily stored in the trunk as previously described [37], while long-term lower levels of FFM were observed compared to the control group. These results contradict those of El Ghoch et al. showing normalization of FFM after weight gain in 90 AN patients [38]. This discrepancy could be explained by the different methods used to measure body composition. El Ghoch et al. used DXA while Hübel included studies with BIA, DXA, dual photon absorptiometry, or magnetic resonance imaging. The restoration of lean body mass is also a key determinant of outcome and quality of life in chronic diseases. Routine investigation of muscle mass is needed in AN patients, particularly to evaluate the impact of new therapeutic tools such as adapted physical activity that has been reported to be associated with increased muscle mass, and to avoid excessive gain of abdominal FM during refeeding [39]. Moreover, AN is an eating disorder associated with body image disturbance and body shape concerns, therefore an inadequate gain of FM can be a major risk factor for relapse, while a gain of muscle mass may enhance a patient’s compliance, therapeutic outcome and long-term recovery. Accurate body composition monitoring should be routinely implemented in the standard care of AN patients during refeeding.

Our study has some limitations. For muscle mass analyses, we evaluated all the equations together although ALM and skeletal muscle mass are not the same. ALM is the sum of the lean soft tissue of the four limbs, while skeletal muscle mass includes ALM, trunk and head skeletal mass. Thus, as discussed before, our results are not comparable with those of other studies using different devices (BIA, DXA) and population characteristics (age, BMI, etc.).

## 5. Conclusions

In conclusion, the routine assessment of body composition is needed for the initial evaluation of nutritional status in AN patients and the monitoring of compartmental weight gain during refeeding. The BIA equations selected here exhibit very high 95% CI highlighting the lack of a disease-specific equation, and the need for qualified clinicians to interpret BIA results in this population. Larger studies are needed to confirm our results. In addition, the validity of BIA equations to longitudinally follow body composition should be further investigated.

## Figures and Tables

**Figure 1 jcm-09-03664-f001:**
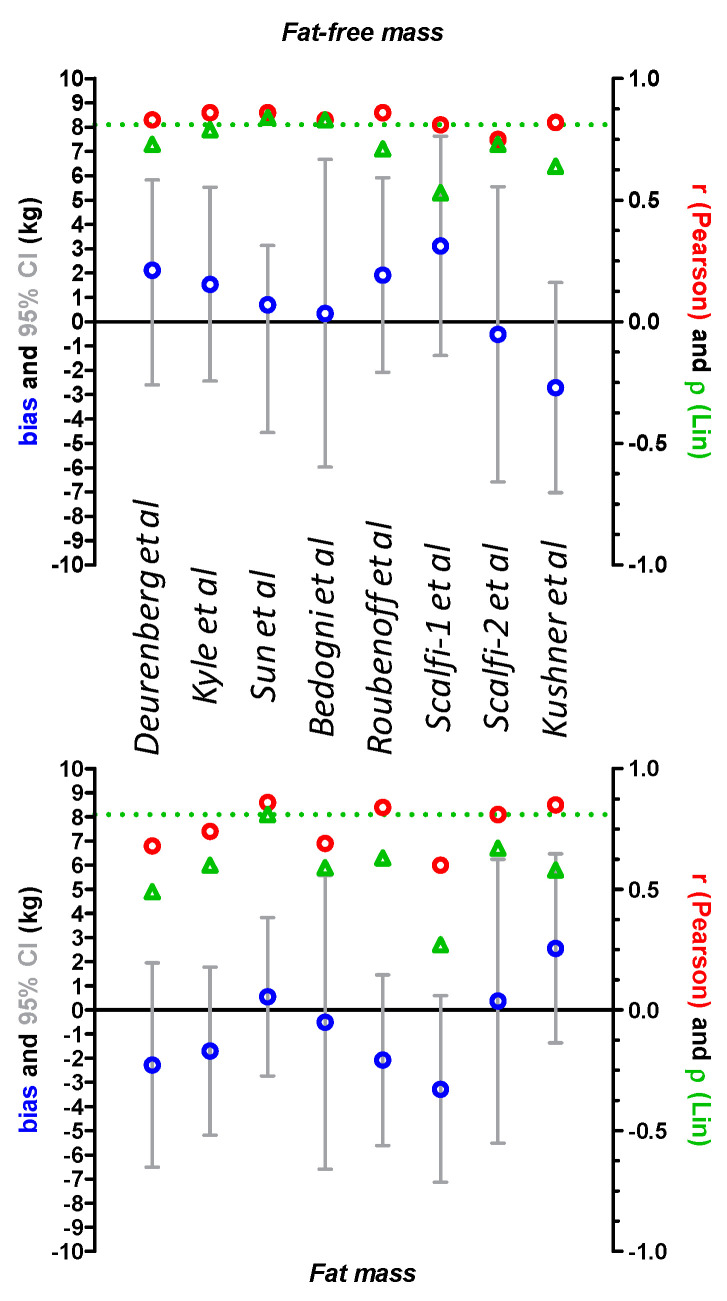
Comparison of fat-free mass (FFM) and fat mass (FM) measured by dual-energy X-ray absorptiometry (DXA) and bioelectrical impedance analysis (BIA) according to BIA equations. FFM (Upper panel) and FM (lower panel) values from 115 severely anorectic patients were compared with a Bland–Altman test. Bias (DXA-BIA values) and 95% confidence intervals (95% CI) are shown in blue and gray, respectively. Pearson correlations were performed and Pearson r is shown in red circles. The concordance coefficient of Lin ρ was calculated and is shown in green triangles. The green dashed line represents the cut-off at 0.81 to consider ρ fairly good.

**Figure 2 jcm-09-03664-f002:**
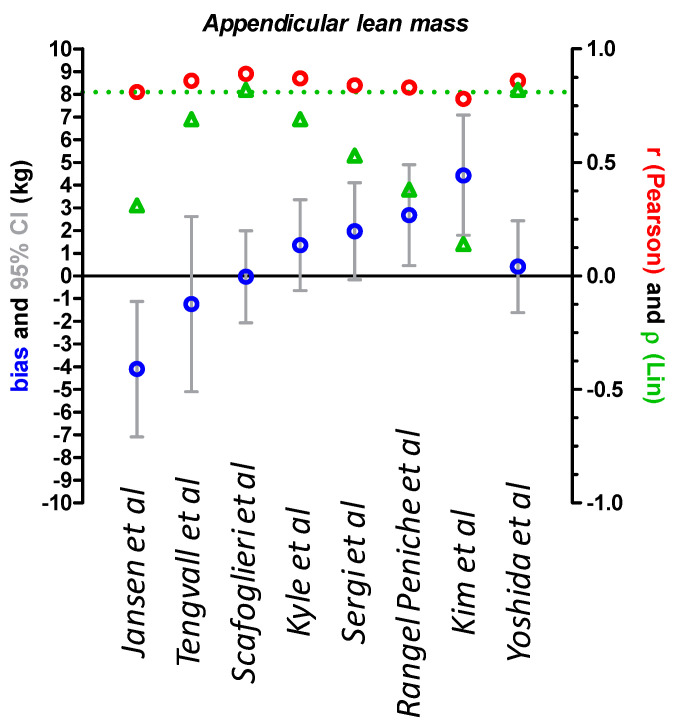
Comparison of appendicular lean mass measured by DXA and BIA according to BIA specific equations. Appendicular lean mass (ALM) values from 115 severely anorectic patients were compared with a Bland–Altman test. Bias (DXA – BIA values) and 95% confidence intervals (95% CI) are shown in blue and gray, respectively. Pearson correlations were performed and Pearson r is shown in red circles. The concordance coefficient of Lin ρ was calculated and is shown in green triangles. The green dashed line represents the cut-off at 0.81 to consider ρ fairly good. DXA: dual X ray absorptiometry; BIA: Bioelectrical impedance analysis.

**Table 1 jcm-09-03664-t001:** Tested bioimpedance equations for body composition.

	Equation
Deurenberg et al. [16]	FFM = −12.44 + (0.34 × height^2^/resistance) + (0.1534 × height) + (0.273 × weight) − (0.127 × age)
Kyle et al. [18]	FFM = −4.104 + (0.518 × height^2^/resistance) + (0.231 × weight) + (0.130 × reactance)
Sun et al. [22]	FFM = −9.529 + 0.696 × (height^2^/resistance) + (0.168 × weight) + (0.016 × resistance)
Bedogni et al. [15]	FFM = 0.6 × (height^2^/impedance 50kHz) + (0.2 × weight) + 3.3
Roubenoff et al. [19]	FFM = 7.7435 + (0.4542 × height^2^/resistance) + (0.1190 × weight) + 0.0455 × reactance
Scalfi-1 et al. [20]	BWa = (0.434 × weight) + 6.326
Scalfi-2 et al. [20]	BWa = (0.563 × height^2^/impedance 100kHz) + 2.695
Kushner et al. [17]	BWa = (8.315 + 0.382 × height^2^/resistance) + 0.105 × weight

BWa, body water; FFM, fat-free mass.

**Table 2 jcm-09-03664-t002:** Tested bioimpedance equations for muscle mass.

	Equation
Janssen et al. [31]	SMM = (height^2^/resistance × 0.401) + (gender × 3.825) + (age × −0.071) + 5.102
Tengvall et al. [26]	SMM = −24.021 + (height × 0.33) + (resistance × −0.031) + (reactance × 0.083) + (gender × −1.58) + (weight × 0.046)
Scafoglieri et al. [25]	ALM = 1.821 + 0,168 × (height^2^/resistance) + (0,132 × weight) + (0,017 × reactance) − (gender × 1.931)
Kyle et al. [23]	ALM = −4.211 × (height^2^/resistance × 0.267) + (weight × 0.095) + (gender × 1.909) +(age × −0.012) + (reactance × 0.058)
Sergi et al. [28]	ALM = −3.964 + 0.227 × (resistance/height^2^) + (0,095 × weight) + (gender × 1.384) + (reactance × 0.064)
Rangel Peniche et al. [29]	ALM = −0.05376 + 0.2394 × (height^2^/resistance) + (2.708 × gender) + (weight × 0.065)
Kim et al. [30]	ALM = ((height^2^/resistance × 0.104) + (−0.050 × age) + (gender × 2.954) + (weight × 0.055)) + 5.663
Yoshida et al. [24]	ALM = 0.221 × (height^2^/resistance) + (0.117 × weight) + 0.881

SMM, skeletal muscle mass; ALM, appendicular lean mass.

**Table 3 jcm-09-03664-t003:** Demographic data of patients.

*n* = 115	Mean ± Standard Deviation	Range
Age (y)	32.3 ± 14.5	18–67
Weight (kg)	39.9 ± 4.6	26.6–50.0
BMI (kg/m^2^)	14.6 ± 1.2	10.9–15.9
ALM (kg)	14.88 ± 2.04	9.35–20.94
FM (kg)	4.2 ± 2.4	1.1–12.5
FM (%)	11.0 ± 5.6	4.0–26.4
FFM (kg)	35.5 ± 3.8	24.2–43.9
FFM (%)	88.2 ± 7.2	73.6–96.0

Body composition was assessed by DXA (Lunar Prodigy Advance). FFM was obtained by adding lean mass and bone mass. ALM, appendicular lean mass; FFM, fat-free mass; FM, fat mass. DXA, Dual-Energy X-ray Absorptiometry. BMI, Body mass index.

**Table 4 jcm-09-03664-t004:** Bioelectrical impedance data.

*n* = 115, Ω	Mean ± Standard Deviation	Range
Reactance	61.31 ± 13.15	29.3–100.4
Resistance	686.2 ± 79.58	517.0–887.0
Impedance 5 kHz	652.6 ± 231.9	30–941
Impedance 50 kHz	689.3 ± 79.82	518–888
Impedance 100 kHz	660.0 ± 77.73	498–856
Impedance 200 kHz	631.0 ± 75.92	473–843

BIA was performed using BodyStat QuadScan 4000. BIA, bioelectrical impedance analysis.

**Table 5 jcm-09-03664-t005:** Comparison of fat mass and fat-free mass values obtained by DXA and BIA according to the BIA equation used.

	Bias (kg)	95% CI (kg)	Bias (%)	Accuracy (%)	Pearson r	Lin ρ
Fat-free mass						
Deurenberg et al.	2.12	(−2.59; 5.83)	5.97	47.8	0.83	0.73
Kyle et al.	1.54	(−2.44; 5.53)	4.33	47.8	0.86	0.79
Sun et al.	0.70	(−4.55; 3.13)	1.97	66.9	0.86	0.84
Bedogni et al.	0.34	(−5.97; 6.67)	0.96	66.1	0.83	0.83
Roubenoff et al.	1.92	(−2.07; 5.92)	5.40	40.0	0.86	0.71
Scalfi-1 et al.	3.11	(−1.39; 7.63)	8.75	22.6	0.81	0.53
Scalfi-2 et al.	−0.52	(−6.59; 5.55)	−1.46	50.4	0.75	0.73
Kushner et al.	−2.71	(−7.03; 1.61)	−7.63	33.9	0.82	0.64
Fat mass						
Deurenberg et al.	−2.28	(−6.51; 1.95)	−54.3	2.6	0.68	0.49
Kyle et al.	−1.70	(−5.19; 1.78)	−40.5	3.5	0.74	0.60
Sun et al.	0.55	(−2.73; 3.83)	13.1	13.9	0.86	0.81
Bedogni et al.	−0.51	(−6.60; 5.58)	−12.1	5.2	0.69	0.59
Roubenoff et al.	−2.08	(−5.61; 1.45)	−49.5	2.6	0.84	0.63
Scalfi-1 et al.	−3.28	(−7.14; 0.59)	−78.1	0.9	0.60	0.27
Scalfi-2 et al.	0.37	(−5.52; 6.25)	8.81	2.6	0.81	0.67
Kushner et al.	2.55	(−1.37; 6.47)	60.7	2.6	0.85	0.58

Bias was calculated by the difference (DXA value-BIA derived value). Accuracy is the percentage of patients with DXA-BIA difference within ±5% of DXA measure. 95% CI, 95% confidence interval. DXA, Dual-Energy X-ray Absorptiometry; BIA, Bioelectrical impedance analysis.

**Table 6 jcm-09-03664-t006:** Comparison of appendicular skeletal muscle mass values obtained by DXA and BIA according to the BIA equation used.

	Bias (kg)	95% CI (kg)	Bias (%)	Accuracy (%)	Pearson r	Lin ρ
Janssen et al.	−4.10	(−7.09; −1.12)	−27.5	1.7	0.81	0.31
Tengvall et al.	−1.24	(−5.10; 2.62)	−8.4	45.2	0.86	0.69
Scafoglieri et al.	−0.03	(−2.06; 1.99)	−0.2	83.5	0.89	0.82
Kyle et al.	1.35	(−0.65; 3.36)	9.0	53.0	0.87	0.69
Sergi et al.	1.97	(−0.17; 4.11)	13.2	30.4	0.84	0.53
Rangel Peniche et al.	2.68	(0.45; 4.90)	18.0	10.4	0.83	0.38
Kim et al.	4.43	(1.79; 7.09)	29.8	0.0	0.78	0.14
Yoshida et al.	0.41	(−1.62; 2.44)	2.8	82.6	0.86	0.82

Bias was calculated by the difference (DXA value – BIA derived value). Accuracy is the percentage of patients with DXA-BIA difference within ±5% of DXA measure. 95% CI, 95% confidence interval. DXA, Dual-Energy X-ray Absorptiometry; BIA, Bioelectrical impedance analysis.

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
