# Peer review of "Validity of Bioimpedance Equations to Evaluate Fat-Free Mass and Muscle Mass in Severely Malnourished Anorectic Patients"

_jcm, 2020, doi:10.3390/jcm9113664_

Round 1
Reviewer 1 Report
The current study sets out to analyze the validity of BIA to predict body composition, with a focus on Fat Free Mass and muscle mass in a reasonably sized group of 115 malnourished adults with Anorexia Nervosa (AN). Limitations in all selected equations were noted, especially high 95% Cis and low accuracy in most equations. The prediction formulas with the relatively highest validity were identified. The study is clearly structured, well written and touches an relevant clinical subject. Measuring body composition before and after weight gain could serve as a marker of successful refeeding programs (i.g. nutritional rehabilitation while allowing for treatment elements enabling gains in reduced muscle mass). However the study would profit from a more thorough research into already existing literature to derive the gaps of knowledge covered with the current study in the introduction, and for a comparison of obtained and previous results in the discussion section.
Minor comments:
- spell check by native English speaker required (e.g. page 1 line 14: “rapid” instead of “rapide”, line 20: kg/m2 instead of kg/m2 or kg.m-2 in line 22; done ON the same day after overnight fasting, is associated with (line 37) instead of associated to – these are just examples.
- please insert age and sex of the study participants into the abstract
- when stating high CI, please either show concrete numbers or else omit the information on CI.
- please use past tense to describe results (line 32: was adapted).
- I would suggest to add “body composition” to the key words
Abstract
Page 14: I have never read the word “Bioimpedancemetry” – is this the correct term? I am familiar with the word “Bioelectrical Impedance Analysis” used by the authors on page 2 line 45 and would recommend keeping to this term.
Introduction
DXA is associated with radiation exposure and therefore in case of young females cannot be considered as non-invasive
Results
- it would be interesting to see minimum and maximum values expressed as range in table one and two
Major comment
There are a range of previous studies on the same topic that are not included in the introduction or discussion, in order to allow to derive the necessity of further studies and a better comparison of the current and previous results, to name and explain similarities or discrepancies of findings.
Examples:
Body composition in subjects with anorexia nervosa: bioelectrical impedance analysis and dual-energy X-ray absorptiometry
G Bonaccorsi 1, A Bassetti, S Chiari, P Dirindelli, C Lorini, C Menicalli, F Santomauro, M G Martinetti
Underweight patients with anorexia nervosa: comparison of bioelectrical impedance analysis using five equations to dual X-ray absorptiometry.
Mattar L, Godart N, Melchior JC, Falissard B, Kolta S, Ringuenet D, Vindreau C, Nordon C, Blanchet C, Pichard C.
The reliability of bioelectrical impedance analysis for measuring changes in the body composition of patients with anorexia nervosa
C L Birmingham 1, P J Jones, C Orphanidou, R Bakan, I G Cleator, E M Goldner, P T Phang
Evaluation of methods to assess reduced body fat in patients with anorexia nervosa
Verena Haas 1, Dorothea Stark, Michael Kohn, Manfred J Müller, Simon Clarke, Caron Blumenthal, Julie Briody, Sloane Madden, Kevin J Gaskin
Author Response
Dear reviewer,
Please see the attachment,
Best regards,
Najate Achamrah

Reviewer 2 Report
Dear authors:
First of all, I would like to thank you for the preparation of this research because body composition is a very important parameter in the recovery of anorectic patients so it is necessary to know the reliability of different techniques available to evaluate it.
Next, I will indicate some aspects that I think would improve the presented article.
- In the abstract, in line 31, I think that the section called “discusión” should be named as a “conclusion”.
- In Materials and methods, It should be indicated if informed consent was signed and if the Lukaski protocol for BIA measurements was followed. Furthermore, it would be convenient to indicate in the text, not only in the figure, that both measurements had to be carried out on the same day. It should be specified if they were carried out in a row or, at least, if the patients were in the same conditions in both measurements.
- Lines 142 and 143 of Results show equations that obtained a Pearson correlation value higher than 0.8, but the Kushner et al equation, which also exceeded this value, is not indicated. Also, in the analysis of muscle mass, all equations are evaluated together but it is necessary to take into account that skeletal muscle mass and appendicular lean mass are not the same so it should not be studied together.
- The thought expressed in line 174 of the discussion does not fully agree with what is shown in Results. Besides, it should be noted that some of the equations were developed in populations with the same pathology. Otherwise, I think that the last paragraph of the discussion is not necessary because it discusses the results of a different technique than the one evaluated in this research. This technique usage could be suitable as possible future research, which has not been indicated, but it is not necessary to compare the results of other authors. Likewise, the limitations of the study have not been indicated.
- It would be necessary to review the bibliography because in some cases journal titles have been abbreviated and in others not. Furthermore, equation references should be numbered as they appear in the tables and reference number 17 does not support any of the equations listed in Table 1.
Kind regards,
Author Response
Dear Reviewer,
Please see the attachment,
Best regards
Najate Achamrah

Round 2
Reviewer 1 Report
The authors have addressed the raised points. They should check the accuracy of the sentence 2 and 3 in the introduction: The assessment of body composition plays a key role in evaluating nutritional status in AN because body weight and BMI could be unreliable in cases of edema for example. Several accurate techniques for the assessment of body composition in human have been developed (3). ..could be unreliable, e.g. in cases of edema... have been developed in humans
This manuscript is a resubmission of an earlier submission. The following is a list of the peer review reports and author responses from that submission.